# Enhancing Aerospace Industry Efficiency and Sustainability: Process Integration and Quality Management in the Context of Industry 4.0

Gheorghe Ioan Pop [1], Aurel Mihail Titu [2,3,*] and Alina Bianca Pop [4,*]

1   S.C. Universal Alloy Corporation Europe S.R.L. Dumbravița 244A, 437145 Maramures, Romania;
    popghitza@gmail.com
2   Industrial Engineering and Management Department, Faculty of Engineering, "Lucian Blaga" University of
    Sibiu, 10 Victoriei Street, 550024 Sibiu, Romania
3   The Academy of Romanian Scientists, 3 Ilfov Street, 030167 Bucharest, Romania
4   Faculty of Engineering, Department of Engineering and Technology Management, Northern University
    Centre of Baia Mare, Technical University of Cluj-Napoca, 62A Victor Babes Street, 430083 Baia Mare, Romania
*   Correspondence: mihail.titu@ulbsibiu.ro (A.M.T.); bianca.bontiu@gmail.com (A.B.P.)

**Abstract:** This paper delves into the multifaceted domain of the aerospace industry, examining its evolution, current challenges, and imperative focus on quality management and process integration. The aerospace sector, driven by technological advancements and a burgeoning global demand for air travel and freight transport, necessitates a thorough analysis of its industrial fabric and operational intricacies. This research endeavors to analyze the dynamics of the aerospace industry, pinpoint its challenges, and propose an integrated approach to enhance efficiency, quality, and sustainability. The primary goals encompass understanding the evolving industry landscape, identifying critical challenges, and offering innovative solutions by amalgamating the principles of Industry 4.0 into quality management and processes within the aerospace sector. Through an in-depth exploration of various facets, this research underscores the pivotal role of efficient processes and integrated quality management in achieving sustainable growth and competitiveness in the aerospace industry. By aligning with the paradigm of Industry 4.0, organizations can optimize their operations and contribute to the industry's advancement, delivering safer and more cost-effective aerospace products. The study adopts a multifaceted approach, incorporating an extensive literature review, a critical analysis of industry trends, the examination of quality management frameworks, and a thorough evaluation of the integration potential of Industry 4.0 technologies. The research also involves case studies and expert insights to validate the proposed approach. The investigation reveals that by leveraging Industry 4.0 technologies and embracing an integrated approach to quality management, the aerospace industry can significantly enhance operational efficiency, product quality, and overall sustainability. The seamless integration of processes and the implementation of advanced quality frameworks pave the way for a more competitive and future-ready aerospace industry, meeting the evolving demands of a globalized world.

**Keywords:** aerospace industry; quality management; process integration; Industry 4.0; efficiency sustainability; technological advancements

## 1. Introduction

In the current industrial landscape, the aerospace and space industries are undergoing a continuous process of evolution and adaptation to support increasingly complex and diverse market requirements. The rapid technological advancement and the growing challenges related to the efficiency and sustainability of products and processes have turned this domain into a competitive and dynamic environment. This evolution has generated

the need for an innovative and integrative approach to quality management and processes in the aerospace industry, aiming to achieve safer, more efficient, and sustainable products.

This study aims to make a significant contribution to the understanding and implementation of advanced quality management practices and process integration in the aerospace industry. In a context where quality and sustainability requirements are becoming more stringent, this research is crucial to optimize processes and ensure that products in the aerospace industry comply with the highest standards of performance and safety.

The current literature reveals gaps in the understanding of the complex processes in the aerospace industry and in the integrated application of quality management in this domain.

While researching the improvements in engineering processes, which form the basis of this paper, we were unable to find studies on process management in the aerospace industry, despite the considerable organizational effort required due to the complexity of requirements.

There are also ambiguities regarding the proper implementation of Industry 4.0 principles in aeronautical and space processes. The main principles of Industry 4.0 considered in these gaps are a revolutionary paradigm that utilizes cutting-edge digital tools, IoT, AI, data analysis, and machine-to-machine communication.

These gaps have motivated the planning and execution of this research to fill the existing knowledge voids and provide new perspectives and innovative solutions.

The primary novelty of this research lies in integrating concepts and technologies from Industry 4.0 into quality management and processes in the aerospace industry. The main concepts implemented are the IoT, data analysis and machine-to-machine communication. The proposed innovative approach will offer a holistic view of quality management and illustrate how it can be adapted and optimized within the specific context of the aerospace industry.

The main goal of this research is to analyze and propose solutions to optimize process integration and quality management in the aerospace industry, using the paradigms and advanced technologies of Industry 4.0. It aims to identify and assess the impact of implementing this innovative approach on the efficiency, quality, and sustainability of products and processes in the aerospace industry.

Process integration, as elucidated in this paper, significantly diminishes the risk of nonconformity, and concurrently reduces costs, enhancing the efficiency of each process, sub-process, and activity, while also promoting sustainability. In conclusion, we believe that by presenting this approach, we provide a valuable point of reference for other systems in this industry.

This paper is organized into eight major parts, each contributing significantly to the understanding and implementation of processes and quality management within the aerospace industry. The first part delves into the evolution of industrial organizations in the aeronautical field, shedding light on the types of organizations involved and current industry challenges. The second focuses on current challenges in the aerospace industry, analyzing the economic and technical aspects. The third part discusses quality management in the aviation industry, emphasizing its crucial role in aircraft design, production, operation, and maintenance. Then, the quality management system in the context of the aeronautical industry, showcasing the importance of adherence to global standards, is explored. The next part uncovers insights into unlocking efficiency and value and current perceptions on process management in the aerospace industry. Then, the modelling systems and processes in the aerospace industry, essential for performance evaluation and improvement, are investigated. The next part provides an overview of the processes and their integration into the quality management system, revealing the interconnections and pivotal role they play. Lastly, the final part presents our conclusions and further research opportunities, offering insights and innovative pathways for future studies.

## 2. Evolution of Industrial Organizations in the Aeronautic Field

The evolution of the industrial environment in the aeronautic domain has significantly impacted the quality requirements for each component of aircraft. Aeronautics, defined as a segment of technology devoted to aircraft manufacturing and air navigation in the Romanian language explanatory dictionary, has undergone substantial transformations throughout its history.

A major consequence of the current complexity of aircraft is the involvement of multiple organizations specialized in various fields of activity, all contributing to a common goal: creating an aircraft that ensures safe flight with minimized costs.

In the aeronautical domain, the following types of organizations are distinguished:

- Design organizations: These organizations develop products (aircraft) according to customer requirements, whether for commercial or military purposes. They are sizable organizations that benefit from modern design technologies, substantially aiding them in product design through simulation of the final product—the aircraft—as well as simulation of the operation of each individual component or system. The simulation and verification methods are dictated by the design standards established by major manufacturers.

- Execution organizations: These are specialized in various activity domains necessary for the completion of the final product. Multiple organizations are required for the realization of the final product, connected through a well-controlled supply chain based on aeronautical domain standards. Figure 1 demonstrates that to achieve an aircraft, suppliers are classified based on levels, according to the configuration of the aircraft components. It also indicates that organizations are structured and specialized in relation to aircraft configuration. This configuration is determined by the aircraft assembly technology, and, at the same time, the execution technologies used. Figure 1 presents the distribution of the Boeing 787 aircraft configuration execution for different suppliers. The aircraft configuration represents the sum of components structured into assemblies, sub-assemblies, and parts.

- Certification organizations: These entities, by their status, can grant various certifications to organizations, attesting their capability to produce specific products or services in aeronautics in compliance with applicable standards. Among the most frequently mentioned certifications in the specialized literature are certifications for the quality management system (e.g., EN AS9100 Quality Management System in Aeronautics [1]), certifications for special production processes (e.g., NADCAP—Aerospace and Defense Contractors Accreditation Program), and certifications by governmental authorities (e.g., EASA—European Union Aviation Safety Agency, FAA—Federal Aviation Administration).

- Aircraft maintenance organizations: These organizations focus on aircraft maintenance, implementing rigorous verification procedures at each stop to ensure compliance with relevant standards. They are crucial in an aircraft's life cycle, ensuring passenger safety through aircraft verification procedures during each stop.

This vast organizational structure, comprising entities specialized in different domains, leads to high aircraft costs. To highlight the significance of addressing this field, we will present some current economic data demonstrating the direct correlation between product complexity (aircraft) and costs, according to the study conducted by Doicin, Rusu, Sokovic and Kopac, in 2008 [2].

Figure 1 illustrates the distribution of the execution of the Boeing 787 aircraft configuration among the different suppliers. The aircraft configuration represents the aggregate of components structured into assemblies, sub-assemblies, and parts.

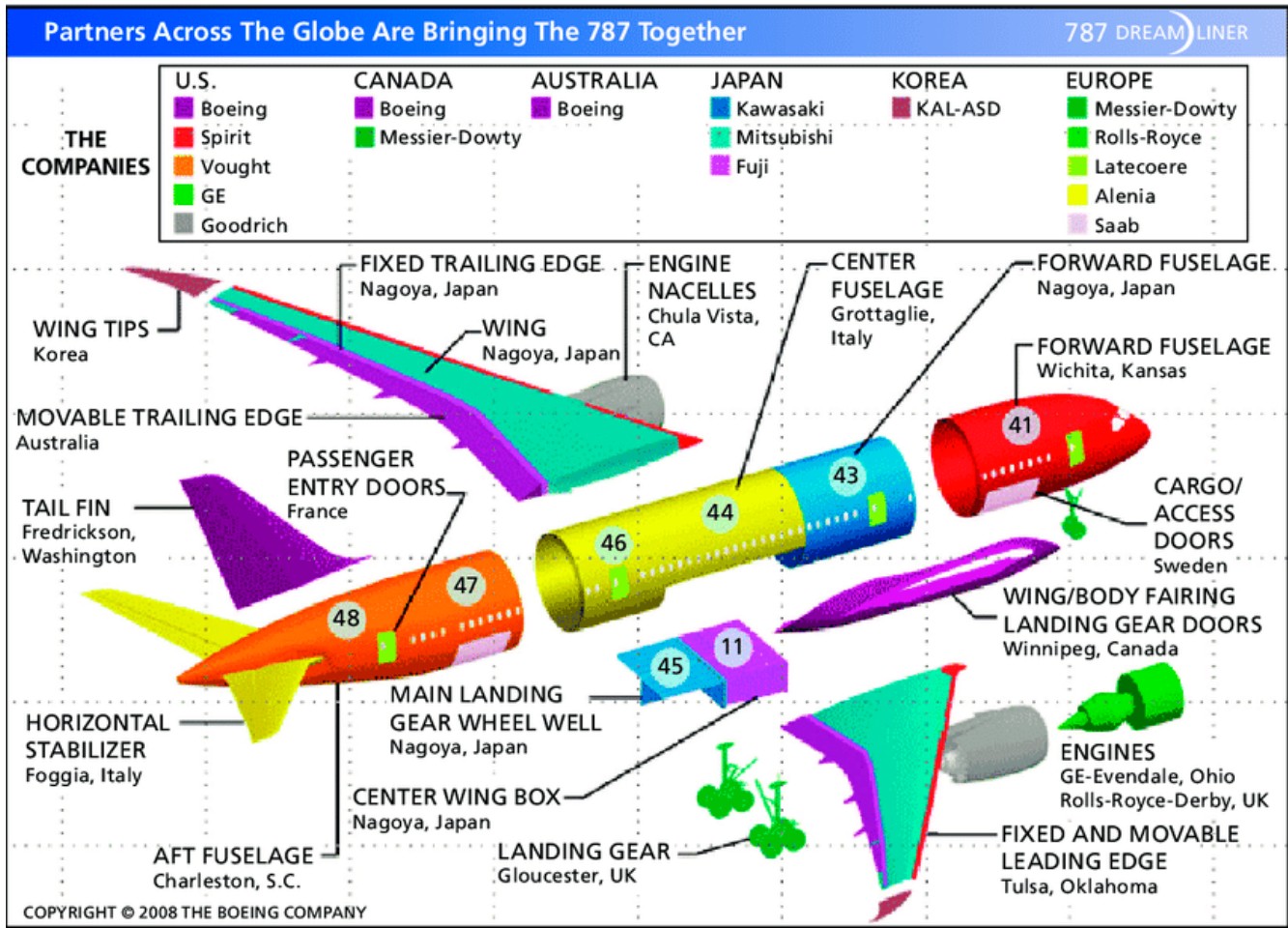

**Figure 1.** Boeing 787 aircraft structure configuration [3].

Addressing the diverse requirements, including product specifications, processes, contractual obligations, certification standards, and legal mandates within the aerospace sector, poses a significant challenge. The complexity of aligning these elements is further heightened by the imperative to ensure timely product delivery.

### 3. Current Challenges in the Aerospace Industry: Analyzing Economic and Technical Aspects

The present aerospace industry is witnessing a continuous surge in air traffic, primarily driven by the expansion of low-cost carriers and the rise of the middle class in emerging economies. This trend has led to an increased demand for new and upgraded aircraft, stimulating investment in research and development.

Simultaneously, the aerospace industry faces significant challenges related to environmental sustainability. Stricter regulations regarding carbon emissions have compelled manufacturers to develop aircraft that are more fuel efficient and explore alternative propulsion options such as electric and hybrid engines.

Efforts in digitization and automation have revolutionized the way aircraft are designed, manufactured, and maintained. Real-time data utilization and Internet of Things (IoT) technology have significantly improved maintenance management, reducing downtime and associated costs.

International collaboration and partnerships among companies from different countries have become essential in the aerospace industry. Sharing expertise and resources contributes to the efficient development of advanced technologies and the standardization of safety norms.

As travel and freight transportation demands continue to grow, the cargo sector has also witnessed substantial expansion. Freight aircraft are optimized for the efficient and rapid transportation of goods, aiding global supply chains.

For aircraft manufacturers and suppliers in the aerospace industry, a major impact comes from the rapid evolution of production technologies. Currently, design organizations collaborate closely with execution and certification teams to innovate and develop new materials and production technologies swiftly to meet the continuously changing market demands. This process is accelerated and facilitated by the emergence of specialized associations in the aerospace domain, promoting common standards and practices, creating a conducive environment for efficient collaboration among organizations with diverse expertise.

Current trends influencing this dynamic include ongoing efforts to reduce the weight of aircraft components to enhance efficiency and autonomy. Concurrently, the development of composite materials and the utilization of 3D printing in manufacturing have opened new horizons in designing and fabricating aircraft components, allowing enhanced flexibility and adaptability in the production process.

As energy efficiency and environmental requirements become increasingly stringent, sustainable technologies have gained critical importance. Investments in low-emission engines, alternative propulsion technologies, and the development of biofuels have become crucial to achieve environmental objectives and maintain long-term competitiveness.

Thus, intensive collaboration among all stakeholders in the aerospace production chain and a commitment to innovation remain fundamental in the face of these challenges and trends. This not only accelerates the development of advanced manufacturing technologies but also ensures that the aerospace industry remains at the forefront of innovation, contributing to the sustainable development of global air transportation.

In the aerospace domain, various methods of transferring documentation from design organizations to execution organizations are being increasingly implemented, utilizing the support of product lifecycle management (PLM) and product data management (PDM) applications.

By leveraging PLM/PDM systems integrated with ERP, aerospace organizations cascade product requirements throughout the entire organizational and process landscape. This approach, utilizing a centralized database for requirements, significantly mitigates the risks associated with using incorrect or missing specifications.

Technical documentation used in the aerospace industry takes various forms [4], including:

- Traditional technical execution documentation (2D drawings, material lists, approvals for document usage in execution).
- Partially digital technical execution documentation (2D drawings with reduced information, 3D models, material lists, approvals for document usage in execution), known as digital product data definitions (DPD).
- Fully digital technical execution documentation (3D models with annotations, material lists, approvals for document usage in execution), known as model-based definitions (MBD).

A significant challenge for aircraft design organizations is to transfer technical information to organizations executing structures, assemblies, components, materials, and infrastructure. Technical documentation aligns with international standards as well as with the standards of major organizations such as Airbus and Boeing, among others. To ensure the proper organization of technical information control, these organizations have developed standardized processes for transferring design data. Having a major impact on product compliance, these processes have been incorporated into international standards for management system certifications.

Figure 2 highlights the way design requirements are transferred from aircraft manufacturers to raw material and component suppliers, in relation to the technological flow for product realization.

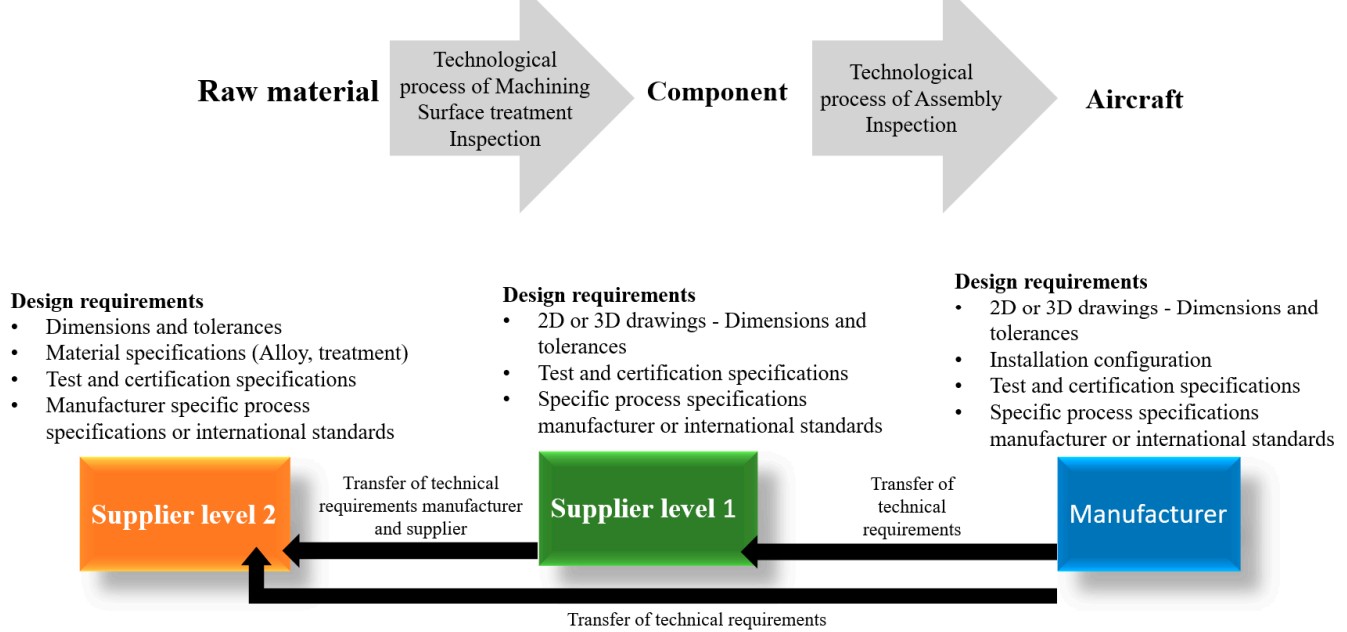

**Figure 2.** Transfer of technical requirements from the final aircraft manufacturer to suppliers.

Using the IDEF methodology in the mind-mapping process, we can identify several critical aspects in the workflow, such as:

- Disconnects in requirements consumption: Process analysis reveals possible areas where the flow of information and requirements may not be smooth or efficient.
- Redundant activities: Identifying and eliminating activities that may be redundant or inefficient within the workflow to increase operational efficiency.
- A lack or excess of control points: Assessing processes to identify if there are too few or too many quality control points to optimize flow and ensure quality.

These detailed analyses provide a robust framework for identifying and addressing potential problems, helping to optimize processes and minimize risk in the aerospace industry.

The aerospace industry has undergone significant evolution in recent years, with one of the primary challenges stemming from the escalating global demand for air travel and freight transport. The demand places immense pressure on aircraft manufacturers and their ability to efficiently manage the expansive worldwide supply chain. Coping with a high manufacturing rate requires industrial organizations, particularly suppliers, to navigate a multitude of requirements.

In response to these challenges, aerospace companies are increasingly looking towards successful approaches employed in other industries, notably the automotive sector, and embracing concepts like Industry 4.0. This shift is driven by the need for more agile, interconnected, and technologically advanced processes. By learning from the experiences of industries that have effectively managed large-scale supply chains and production demands, the aerospace sector aims to enhance its adaptability and responsiveness to the evolving landscape of global air travel and freight requirements.

## 4. Quality Management in the Aviation Industry

Quality management projects share fundamental quality principles, tailored, and applied uniquely across various industrial domains. In the aerospace industry, quality is crucial in aircraft design, production, operation, and maintenance, given their long lifespan and safety requirements. The AS9100 standard, developed by the International Aerospace Quality Group (IAQG), serves as the global benchmark for this industry. The aerospace and defense industry faces strict quality and safety requirements, operating

in complex environments with trends toward shorter production cycles. Investments in the quality management system, aligned with the standards, are vital to meet customer needs and ensure compliance with global requirements. AS9100 [1] is widely adopted and strongly endorsed by major manufacturers, representing a comprehensive quality management system that provides guidance for the efficient implementation of standards in the aerospace industry [5,6] (Figure 3).

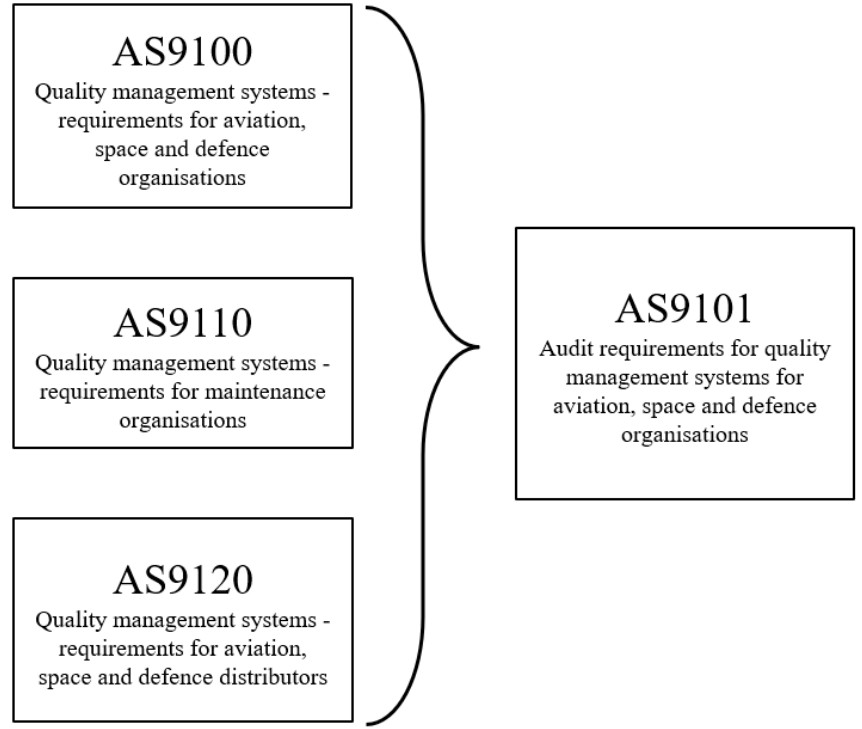

**Figure 3.** Certification scheme for aviation organizations [1,7–9].

Adopting a process-based quality approach is essential in the aerospace industry. It involves systematically defining and managing processes, focusing on their interactions to achieve the desired outcomes in alignment with the organization's quality policy [10,11]. This approach helps in understanding and consistently meeting requirements, adding value to processes, achieving process performance, and improvements based on data and information evaluation. A cohesive system of interconnected processes is crucial for organizational efficiency and effectiveness in achieving its objectives.

Continuous improvement is a cornerstone for successful organizations. It entails enhancing products and services to meet current and future needs, preventing, or reducing adverse effects, and improving the performance and effectiveness of the quality management system. The plan–do–check–act (PDCA) methodology is a valuable tool for implementing continuous improvement, involving stages of planning, implementation, monitoring, and taking corrective actions. This iterative cycle drives the maintenance and enhancement of the quality management system [12].

## 5. Quality Management Systems in the Context of the Aeronautical Industry

In the aerospace industry, integrating legal requirements into the quality management system is of paramount importance due to the critical nature and safety of the products and services offered. This process involves the following essential aspects:

- *Standards and international agencies:* Regulations in the aerospace domain are developed and verified by international organizations such as ICAO, CAA, FAA, and EASA. These standards serve as references for states in developing their national regula-

tions, harmonizing regulations globally, and ensuring an adequate level of security and efficiency.

- *Product traceability:* Traceability is a crucial aspect in the aerospace industry, allowing the identification of the origin and history of each item. It involves marking products with essential information such as part code, production batch, and other relevant details. This information is vital for identifying processes and materials used in case of non-conformities or maintenance checks.
- *Auditing in the aerospace industry:* Auditing is a systematic, independent, and documented process used to objectively evaluate compliance with legal requirements. In the aerospace industry, audits are not limited to ensuring compliance but also aim to identify weak points in the management system. These audits are essential to provide feedback and contribute to the continuous improvement of systems.
- *Advanced product quality planning (APQP):* Adapted from the automotive industry, APQP is essential in the aerospace industry to manage the complex requirements of products. This process involves planning and implementing quality assurance measures from the product conception phase to production and post-production. It is a mandatory requirement for suppliers in the aerospace industry.

The correct and efficient integration of these aspects into the quality management system in the aerospace industry is essential to ensure compliance with regulations, guarantee the safety and reliability of products, and enable continuous improvements in this critical domain.

## 6. Unlocking Efficiency and Value: Current Perceptions on Process Management in the Aerospace Industry

### 6.1. Processes: The Beating Heart of Organizations

In the dynamic realm of organizational functioning, processes stand as vital entities. Michael Hammer and James Champy, pioneering in the 1990s, emphasized that organizations could significantly enhance productivity by redesigning their business processes [13]. This enhancement translates into faster deliveries, shorter order-to-cash cycles, and workforce savings. However, many organizations lack formalized procedures, relying on historical ways of doing things. Despite this, analyzing or improving these processes becomes challenging without explicit process descriptions.

### 6.2. Unveiling the Inefficiencies: Breaking the Mold

In many cases, organizations continue redundant tasks for decades without questioning their purpose. Risks, such as unnecessarily routing a document to multiple individuals, lurk in such scenarios, resulting in waiting times for additional approvals. When asked why things are done this way, the common response is, "That's how we've always done it". This phenomenon stems from organizational inertia, where questioning established practices rarely occurs. Akhil Kumar aptly asserts [14]:

- A process is a sequence of activities aiming to achieve an objective.
- A process model is a formal representation of related activities progressing in a specific order to achieve a clear objective.
- A business process involves activities that require one or more input types, producing an output valuable to the customer.
- A business process is defined as a chain of activities ultimately producing a specific product for a specific customer or market.

### 6.3. Dividing and Conquering: The Process Spectrum

Within organizations, there are various types of processes: management processes, operational processes, and support processes.

### 6.4. Management Processes: Guiding the Ship

Management processes follow a six-stage lifecycle proposed by Dumas and colleagues, encompassing identification, discovery, analysis, redesign, implementation, and monitoring and control [15]. Planning, organizing, coordinating, and controlling form the four functions of management, operating as a continuous process. Planning sets the organization's objective and decides on the best course of action to achieve it, essentially determining the organization's current and future positions.

### 6.5. Organizing: Turning Plans into Action

Once objectives and plans are established, the next managerial function is the organizing. This involves arranging the human and other resources identified during planning to achieve the objective. It entails determining how activities and resources are combined and coordinated, aiming to create an environment for optimal human performance [16].

### 6.6. Leadership: Paving the Way

Leadership, the third fundamental managerial function, involves influencing and mobilizing people toward a specific goal or direction. Considered the most challenging and impactful of all managerial activities, leadership entails creating a positive attitude toward work and organizational goals among members [17].

### 6.7. Control: Steering towards Success

Control is the monitoring of organizational progress toward achieving objectives. It involves measuring performance, comparing it against existing standards, and finding and correcting deviations. Monitoring progress is crucial to ensure the organization achieves its set goals.

In conclusion, the management process functions are interconnected and cannot be disregarded. Management projects and maintains an environment where personnel, working together in groups, efficiently achieve selected objectives [18].

### 6.8. Operational Processes: The Manufacturing Symphony

Operational processes in an industrial organization coordinate activities to create products. These processes represent a sequence of activities performed with the aid of equipment and natural processes, organized, and guided by people to obtain products. Systemically, the objectives of the production process involve transforming inputs (materials, labor, energy, etc.), into outputs in the form of semifinished goods, finished products, or services [19].

### 6.9. Support Processes: The Unsung Heroes

The intense focus on improving operational processes has somewhat diminished the attention on support processes, which nevertheless exert a direct influence on them and, consequently, on the products. Every organization comprises main, management, and support processes:

Main processes relate to production and services provided.

Management processes pertain to policy setting, organizational objective analysis, resource analysis, and decision-making [20].

Support processes encompass human resources, infrastructure, the work environment, provisioning, transportation, logistics, internal audits, improvements, and more.

### 6.10. Empowering Business Processes: The Technological Edge

Supporting business processes through methods, techniques, and software applications aims to design, adopt, control, and analyze operational processes involving people, organizations, applications, documents, and other sources of information.

In the ever-evolving landscape of the aerospace industry, mastering these processes is pivotal for growth, innovation, and ultimately, soaring to new heights. The current

perceptions outlined here offer a compass for organizations to navigate this dynamic journey, ensuring they stay ahead in the race to excellence [21].

## 7. Modelling Systems and Processes in the Aerospace Industry

In the captivating universe of the aerospace industry, modeling systems and processes is the key to performance. Each process is evaluated based on its outputs and its integration within the organizational structure. As W. Edwards Deming once said, process maps are like a map of uncharted territory—essential for navigating the world of business successfully [22].

The process map serves a pivotal role in the aerospace industry, acting as a critical tool within quality management system (QMS) documents and value stream maps (VSM). In QMS documents such as procedures and work instructions, the process map provides employees with a visual representation of their workflow. This visual aid is invaluable during audits, enabling auditors to gain a comprehensive understanding of the process. Additionally, VSM maps play a crucial role in project planning, offering insights into workflow dynamics and providing essential timeline information.

Process maps provide a visual perspective on the relationships and dependencies between processes. They are the cornerstone for effectively managing business processes, allowing for an understanding of how the organization operates without getting lost in the details. Designing these maps is both an art and a science, crucial for the success of business process management.

In this labyrinth of processes, a well-designed architecture is like a compass. Organizing and documenting processes become essential to ensure compliance and to illustrate the principles that govern the organization's functioning. A well-crafted process map is the beacon that lights the way in times of change or when explaining how the organization operates [23–26].

As can be seen in Figure 4, in an "N" process, all process inputs are transformed by performing coordinated activities against a management thinking, resulting in value-added outputs. Each influencing factor has a greater or lesser impact on the outputs.

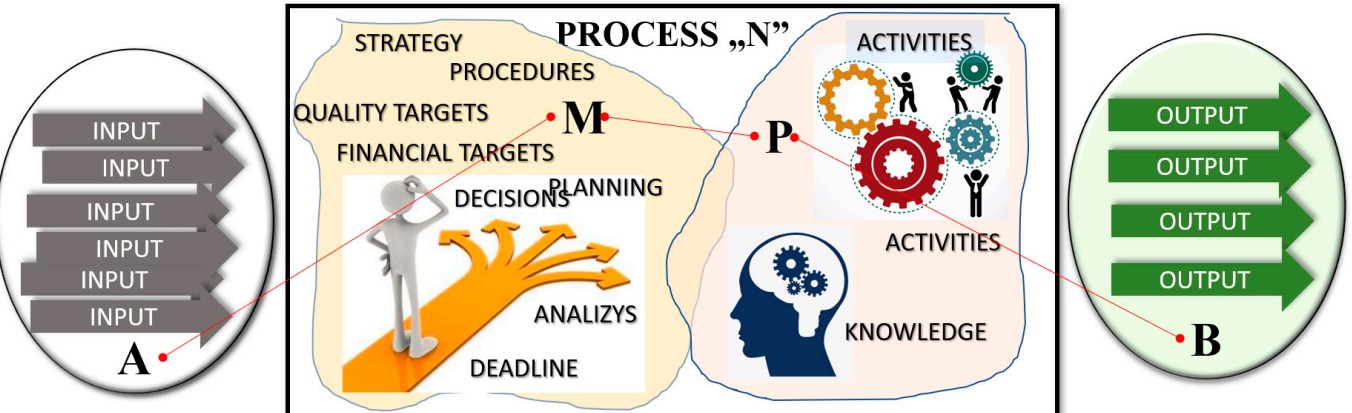

**Figure 4.** Influencing factors in a process.

Each process within the organization has an efficiency calculated in relation to the quality and quantity of the outputs of that process, without considering the quality and quantity of the information input to the process. The influence of management activities (Figure 4 (M)) within processes is achieved by organizing the activities within the processes using procedures and work instructions (Figure 4 (P)). These procedures and instructions are developed based on the structure of the quality management system, the context of the organization and the knowledge of the process owner. Thus, the activities within each process are influenced by the quality management system. The level of influence on the process also indicates the level of integration of the process into the organization's system.

It can be stated that, now, in all industrial organizations, the assessment of the integration of processes into the quality management system is carried out by conducting process audits using the quality management system standard as a criterion.

In conclusion, in our journey through this complex landscape, we turn our attention to engineering processes—the heart and soul of the aerospace industry. Here, experience and challenges have taught us that perfection and innovation stem from modeling and understanding. This is the path to the sky, and we are the guides on this fascinating journey.

In the industrial aerospace sector, engineering processes are vital for translating product quality requirements into production mandates. They are the linchpin of organizations, directly influencing product production and quality. These processes encompass product design, execution documentation, configuration of technical data systems, technological process design, and quality engineering [27].

Product design processes are pivotal, aiming to utilize technical knowledge to develop customer-requested products. They are fundamental in organizations focused on product design and involve designing the necessary technology for product manufacturing and equipment devices [28].

The execution of documentation creation processes involves converting designed product documentation into organization-specific production documentation. This includes technical execution drawings, visual aids for product shape and dimensions, installation/uninstallation files, and programs for various processes [29–31].

Configuration processes of technical data systems establish and control the flow of product technical data within organizations, crucial for production processes. In the aerospace domain, emphasis is on managing technical data, vital for production and product qualification.

Technological process design processes convert designed product and process requirements into a production technological flow, considering process and production requirements [32,33].

The global engineering process, a set of interconnected engineering processes, plays a pivotal role in industrial aerospace organizations. It encompasses product design, production documentation creation, configuration of technical data systems, technological process design, and quality engineering processes.

In the aerospace industry, each product is identified by a set of documents containing design data and legal requirements related to product safety. Design challenges arise from the need for lightweight products with high resistance to various stresses.

To proactively improve product quality, it is crucial to examine and analyze the current process, derive conclusions, and hypothesize improved approaches. The process begins with analyzing product documentation, ensuring completeness and correctness.

Recording and structuring technical data in a database is essential, facilitated by modern applications like PLM, PDM, or DMS. Managing this data is vital for efficient change management, especially given the multitude of information and the need for quick adaptations.

Engineering processes are pivotal in any production organization, bridging the gap between design and production. They transform technical quality requirements into production mandates, considering design, production, and quality specifications.

In the aerospace industry, the global engineering process involves analyzing product documentation, structuring technical data, and efficiently managing this data in a digital environment. This process is essential for ensuring product quality and efficiency in the production of aerospace components and structures.

The global engineering process in the aerospace industry is a captivating journey where technology, precision, and innovation come together to create complex and safe products. Technical documentation databases become vital hubs, connecting multiple departments, and ensuring an efficient flow of information within the organization [34–39].

Products come to life from the designed requirements, and the 3D model becomes their pulse, incorporating essential geometric requirements for machining and inspec-

tion. The evolution towards virtual models has revolutionized production and inspection, eliminating human errors and ensuring quality right from the production phase (Figure 5).

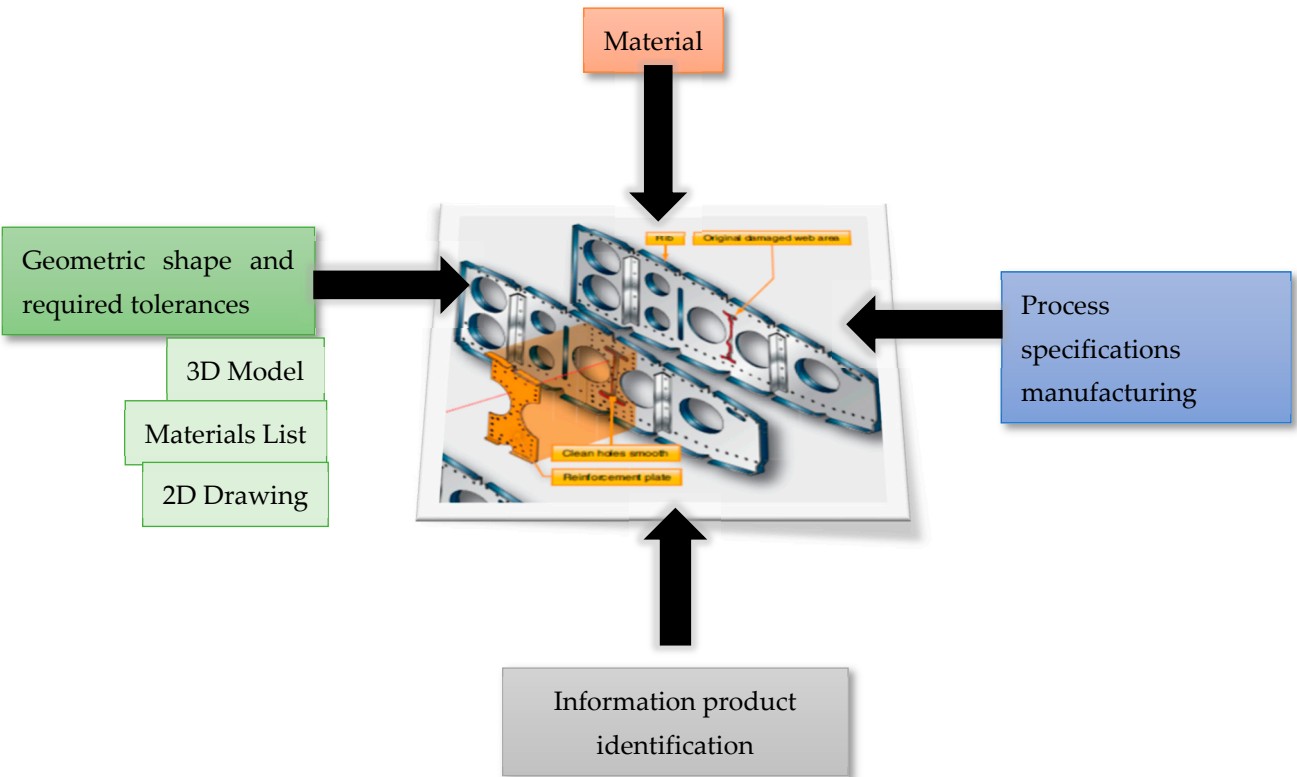

**Figure 5.** Requirements for structural products in the aircraft industry.

From coordinate measuring machines to computer-assisted machining strategies, the aerospace industry continually adapts to achieve high-quality products. Using the same geometric reference throughout the stages—from machining to inspection—ensures that products reach the desired quality before reaching the client.

The global engineering process is a complex orchestration, where CNC programming and device design become keys to efficient production. In an industry where details matter immensely, simulations become imperative to minimize risks and ensure precision (Figure 6).

Behind every final product lies a series of critical decisions, from choosing the semi-finished products to defining the technological itinerary. Internal inspections and tests guarantee quality throughout the process, and inspection plans become quality control roadmaps.

In an environment where safety and precision are non-negotiable, the global engineering process in the aerospace industry represents a dance between technology, strategy, and execution, where every move must be precise and perfectly orchestrated to create products that exceed expectations.

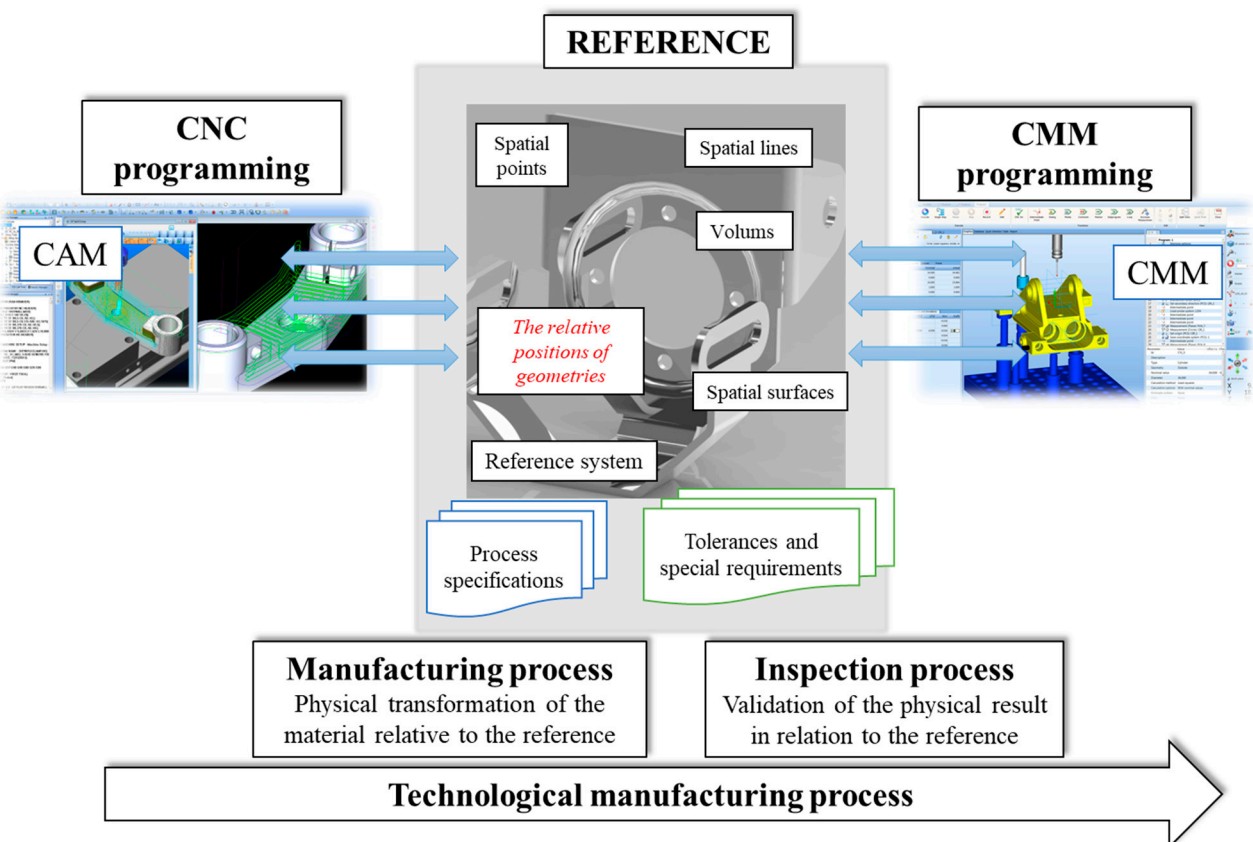

**Figure 6.** Using 3D modelling in production processes.

## 8. Processes and Their Integration into the Quality Management System

Figure 7 shows the distribution and connections of processes in an industrial organization in the aeronautical sector, with a focus on the manufacture of structural metal components. The processes are distributed according to their purpose within the organization. Thus, it is possible to identify:

- Management processes;
- Operational processes;
- Support processes.

As can be seen in Figure 7, customer requirements are the main input to the organizational system. These requirements are transformed by the organization through the organizational context. The outputs, the resulting products, verified by the customer, are also measured, besides the financial benefits, by customer satisfaction, which has a direct impact on the performance evaluation of the organization and on the continuous improvement process.

Leadership is a critical management process in organizations, heavily focused on top management engagement and involvement in processes. Commitment is demonstrated by taking responsibility for quality management effectiveness, integrating quality goals with organizational context, allocating resources for the quality management system, and promoting continuous improvement.

Quality policies in aerospace include adherence to laws and aviation regulations, clear quality objectives, data-driven decision-making, risk-aware thinking, and active improvement of the quality management system.

Planning is fundamental to organizational success, seen at all hierarchical levels. It involves projects for development, production, and improvement, aligning with quality objectives. Coordinating planned activities is vital, analyzing their impact and influence on the organizational system.

Control processes involve evaluating organizational performance through management analyses, audits, and a focus on continuous improvement. Customer satisfaction serves as a valuable marketing tool and influences product quality and improvement processes.

Support processes manage human resources, covering hiring, competence evaluation, awareness, and communication. Quality of human resource knowledge significantly impacts the entire organization.

Operational processes are the organization's "engine". Operational planning and control, crucial sub-processes, develop and monitor operational plans, aligning with customer requirements to ensure product compliance and success.

The operational processes include the sub-processes of product and service design and development, which also have a major impact on the achievement of the operational plan. The production sub-processes bring direct value to the organization. In the organization chosen for this paper, production processes represent the complete technological flow of the production of structural components (Figure 8). Production processes require direct or indirect control. In the aircraft industry, production processes are controlled by the major aircraft manufacturers by defining the production process and inspection standards that suppliers must meet in the technological process of component production.

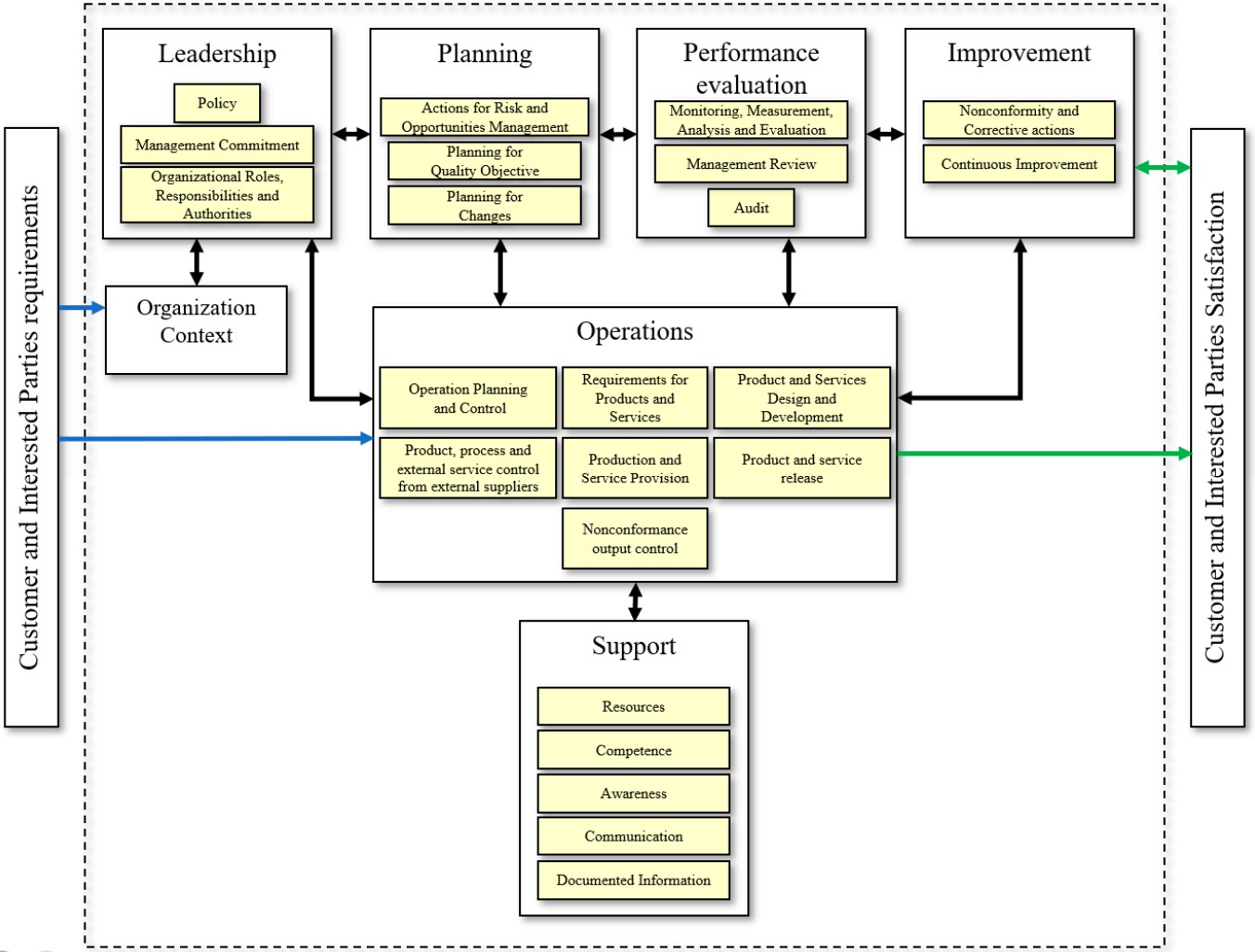

**Figure 7.** Process map of an industrial organization in the aeronautical industry.

**The production processes applicable to the realization of metallic structural products in the aeronautical field**

**Figure 8.** Production processes in industrial organization.

These process and inspection standards are called process specifications and are intended to set the same production process parameters and their limits for all suppliers in the industry. In this way, major manufacturers can design and plan the production process across the entire supply chain to achieve quality products. Controlling the process parameters by various methods is a direct control of the production process. In other words, the resulting process parameter values are directly monitored within pre-defined limits. Process control can also be achieved indirectly by monitoring certain product characteristics resulting from the process under analysis.

In the aeronautical sector, certain production or inspection processes are of a more special nature and are considered to have a major impact on products. For this reason, aircraft manufacturers are directly involved in the qualification of suppliers, conducting qualification and process assessment audits. Thus, material/semi-finished product manufacturing processes, material/semi-finished product heat treatment and inspection processes, certain mechanical machining processes, chemical surface treatment processes, non-destructive inspections, assembly processes and assembly element inspections are considered as special (Figure 9).

As shown in Figure 10, product inspection can be carried out similarly to process inspection, directly or indirectly. Direct product control is achieved by comparing the designed requirements with values or attributes measured directly on the product. Indirect control is achieved by controlling process parameters applied to the product in question.

The process of controlling the resulting non-conformities is critical and is applied at all hierarchical levels because of the potential impact of non-conformities, in terms of safety, which is critical in aviation, and of course in terms of cost. Thus, most organizations in the aeronautical sector have policies to manage non-conforming products under special conditions, by very clearly identifying and segregating them from production products. Also, products declared as scrap are destroyed immediately.

In this paper, the integration of CAD/CAM and CMM processes into the quality management system (QMS) is presented as a pivotal step towards enhancing quality, efficiency, and sustainability in the aerospace sector. The interconnected nature of these systems, facilitated by 3D models, offers several advantages:

- Quality: Mitigating the risk associated with the transfer of design data is crucial. The integration ensures a seamless flow from design authority to NC programming and inspection processes, thereby reducing errors and enhancing overall product quality.

- Efficiency: Consolidating all requirements in a unified system streamlines processes and enables rapid adaptability to changes. This integrated approach promotes efficiency by allowing for quick and coordinated responses to evolving project needs.
- Sustainability: The adoption of virtual environments and the optimization of resources contribute to sustainability. This approach minimizes the physical resources required, aligning with environmentally conscious practices and promoting long-term sustainability in aerospace manufacturing.

By leveraging these advantages, the aerospace sector can not only meet the increasing demands for quality and efficiency but also align with global initiatives for sustainable and responsible industrial practices.

The integration of processes and various organizational systems consistently reduces the risk of overlooking product requirements. Moreover, the communication of new requirements or changes in requirements across the entire organization becomes faster and more effectively controlled. This applies to diverse requirements, whether they are related to products, processes, systems, certifications, or legal standards.

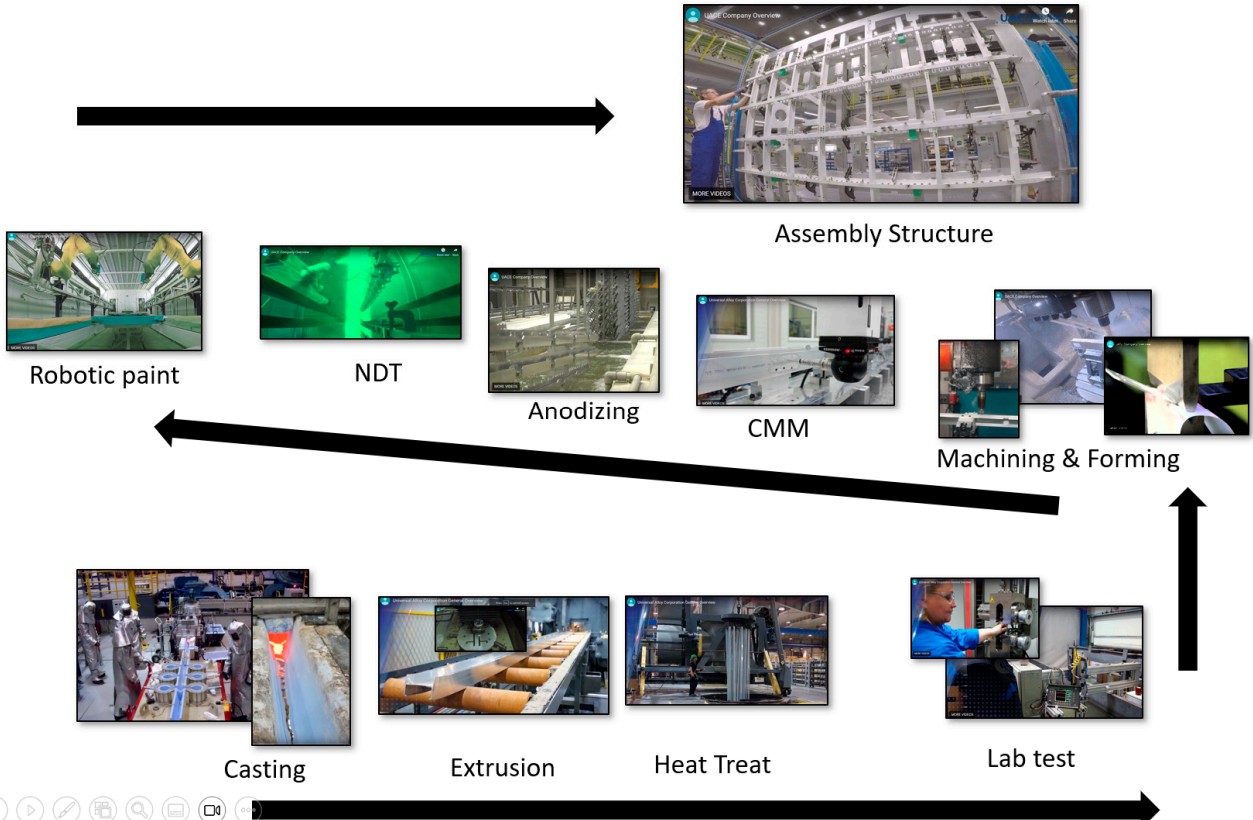

**Figure 9.** Technological process of achieving structural milestones in the aeronautical industry (https://www.universalalloy.com/).

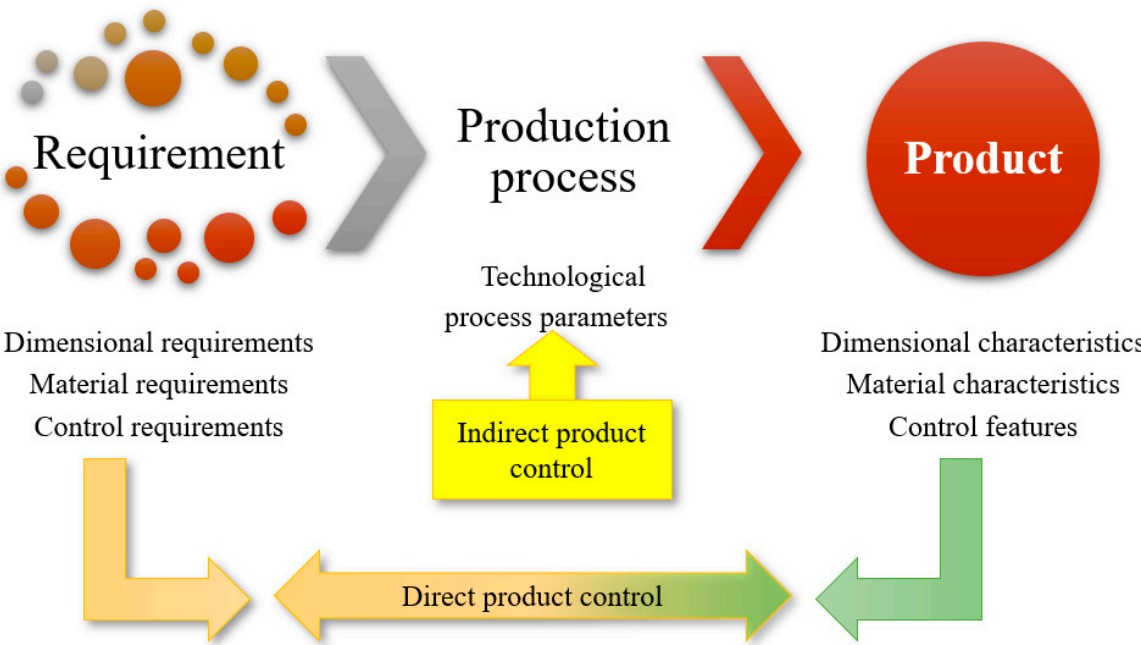

**Figure 10.** Direct or indirect control of the product.

### 9. Conclusions and Further Research

The primary objective of this research is to propose and explore an engineering processes improvement approach within the aerospace industry by incorporating key concepts from Industry 4.0. This entails leveraging advanced technologies, data-driven decision-making, and interconnected systems to enhance the efficiency, quality, and sustainability of engineering processes.

By embracing Industry 4.0 principles, the research aims to streamline operations, optimize resource utilization, and minimize environmental impact. The integration of smart technologies and data analytics is anticipated to lead to more precise and efficient engineering processes, ultimately contributing to the overarching goals of improved operational efficiency, elevated product quality, and a heightened focus on sustainability within the aerospace sector.

To handle the intricate demands of Industry 4.0, aerospace industry organizations are integrating and adjusting to current systems. Technological innovation, competitiveness, and product quality all have an impact on the evolution of industry. Organizational success in the information age is contingent upon their capacity to acquire, develop, and modernize their technology proficiencies.

The 21st century is characterized by innovation and change, which makes it necessary to spread information to reduce risks. To create safe and affordable airplanes, the aerospace industry places a strong emphasis on process integration and quality. By streamlining procedures and requirements, an integrated management system improves organizational performance and creates value.

In the intricate world of management, foresight and insight serve as essential building blocks for success. A great manager sees possibilities and anticipates future developments in addition to providing leadership. Effective management is built on the foundation of this capacity to see the big picture and identify emerging trends.

In essence, it is impossible to see the managerial process as a patchwork of distinct roles. It is a cohesive system in which creating and preserving a supportive work environment is crucial. It is necessary to set up and improve this environment in order to accomplish goals as effectively as possible.

And how do you imagine and create this setting? This is the role of the process map. It is more than simply an example; it is an effective instrument that can change how

businesses perceive and carry out their daily business. The process map becomes a strategic weapon, especially in the aerospace sector where quality criteria are critical.

This map is more than a simple depiction. It is an intricate panorama that shows value flow and interdependencies. It assists companies in broadly allocating their duties while adhering to the core values of quality management. It serves as a map of accountability and integration that explains how procedures affect the production of value.

In the aerospace industry, where control and accuracy are essential, the quality management system is customized to meet the unique needs of the clients. For example, Airbus has developed its own supplier management system to guarantee the caliber of the goods they procure. Customization and adaptability are therefore essential in satisfying clients' ever-more-complex needs and expectations.

However, putting these principles into strategic practice can be challenging. Distinct methods for characterizing and comprehending processes may result in dysfunctions. A functional system depends on striking a careful balance between efficiency and intricacy.

The technical flow is at the center of this maze-like web of procedures, directly affecting support and management procedures. A thorough examination of these linkages leads to exciting new areas of study. There are several areas of study that might be explored, including process mapping, creating effective flows, and identifying places that pose significant hazards.

These days, technology offers strong instruments. Evolution is accelerating from Industry 4.0 to the developing idea of Industry 5.0. But humans continue to be at the core of this progress. While technology can aid, advancement is mostly driven by human ingenuity and expertise.

In conclusion, new and exciting study paths have been revealed because of this exploration of processes and management. Challenges and possibilities include evaluating integration, shortening the industrialization period, and using human expertise. The achievement of striking the ideal balance between technological advancement and human creativity will be crucial as we approach potentially revolutionary shifts.

**Author Contributions:** Conceptualization, G.I.P. and A.M.T.; methodology, G.I.P. and A.M.T.; software, G.I.P.; validation, A.M.T.; formal analysis, A.B.P.; investigation, G.I.P. and A.M.T.; resources, G.I.P.; data curation, G.I.P. and A.M.T.; writing—original draft preparation, G.I.P., A.M.T. and A.B.P.; writing—review and editing, G.I.P., A.B.P. and A.M.T.; visualization, A.M.T. and A.B.P.; supervision, A.M.T.; project administration, A.M.T. and G.I.P. All authors have read and agreed to the published version of the manuscript.

**Funding:** This research received no external funding.

**Data Availability Statement:** Data are contained within the article.

**Conflicts of Interest:** Author Gheorghe Ioan Pop is employed by the company S.C. Universal Alloy Corporation Europe S.R.L. Dumbravița 244A. The remaining authors declare that the research was conducted in the absence of any commercial or financial relationships that could be construed as a potential conflict of interest.

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
