# Peer review of "Enhancing Aerospace Industry Efficiency and Sustainability: Process Integration and Quality Management in the Context of Industry 4.0"

_sustainability, doi:10.3390/su152316206_

Round 1
Reviewer 1 Report
Comments and Suggestions for Authors
Overall, the study was great and well-written. The number of references cited in this article is optimal for this research. The topic of the study is current, and the methodology used in this research is appropriate.
Reviewer 2 Report
Comments and Suggestions for Authors
This paper analyzes the dynamics of the aerospace industry, pinpoints its challenges, and proposes an integrated approach to enhance efficiency, quality, and sustainability. The topic is meaningful but the work is somewhat flawed.
1. According to the introduction of this paper, the concept and technology of Industry 4.0 will be integrated into the quality management and process of the aerospace industry, but the whole article does not explain and define the concept of Industry 4.0.
2. According to the introduction of this paper, there are many defects in the existing literatures. Why don't list the literatures and the specific defects?
3. Process integration and quality management in the aerospace industry are optimized in this paper, but it does not compare with the previous methods and does not quantify the indicators. How can you prove that the method you proposed has optimized process integration and quality management?
4. The conclusions of this paper are not clear in their description of the main contributions of this paper.
5. Check the correctness of the figure number “1.9” on line 110.
Comments on the Quality of English Language
The language is OK
Reviewer 3 Report
Comments and Suggestions for Authors
The manuscript is well-organized and can be recommended for publication. In the meantime, addressing the following question in the revised manuscript can help the reader and clarify the issues better:
1-How has the aerospace industry evolved in recent years, and what are the current challenges it faces, especially in the context of increasing global demand for air travel and freight transport?
2-What are the primary goals of this research in the aerospace industry, and how does it aim to enhance efficiency, quality, and sustainability?
3-How does the integration of Industry 4.0 principles into quality management and processes benefit the aerospace sector, and what specific advantages can be expected from this integration?
4-Could you provide examples of case studies or expert insights that validate the proposed approach in the research?
5-What role does the process map play in the aerospace industry, and how does it contribute to understanding interdependencies, the flow of value, and meeting quality requirements?
6-How do organizations in the aerospace industry adapt and integrate modern systems to manage the complex requirements of Industry 4.0, and what impact does this have on product quality, competition, and technological advancement?
7-In what ways is an integrated management system valuable in optimizing organizational performance in the aerospace industry, and how does it address quality and process integration?
8-What are the key challenges and considerations when customizing and adapting quality management systems to meet the complex expectations and requirements of aerospace customers?
9-What research possibilities are suggested by the technological flow within the aerospace industry, especially in terms of process mapping, efficient flows, and identifying areas with major risks?
10-How does the balance between detail and efficiency affect the functional system of the aerospace industry, and what are the implications of different approaches in defining and understanding processes?
Round 2
Reviewer 3 Report
Comments and Suggestions for Authors
The revision is acceptable